# Effect of Pre-Corrected pH on the Carbohydrate Hydrolysis of Bamboo during Hydrothermal Pretreatment

**DOI:** 10.3390/polym12030612

**Published:** 2020-03-07

**Authors:** Lingzhi Huang, Zeguang Yang, Mei Li, Zhaomeng Liu, Chengrong Qin, Shuangxi Nie, Shuangquan Yao

**Affiliations:** 1School of Light Industrial and Food Engineering, Guangxi University, Nanning 530004, China; huanglingzhi9801@163.com (L.H.); 18889200554@sina.cn (Z.Y.); Limei991010@163.com (M.L.); zhaomengliu@163.com (Z.L.); qcr2019@sina.com (C.Q.); nieshuangxi@gxu.edu.cn (S.N.); 2Guangxi Key Laboratory of Clean Pulp & Papermaking and Pollution Control, Nanning 530004, China

**Keywords:** hydrothermal pretreatment, pre-corrected pH, hemicellulose, cellulose, lignin

## Abstract

To confirm the prospects for application of pre-corrected pH hydrothermal pretreatment in biorefineries, the effects of pH on the dissolution and degradation efficiency of carbohydrates were studied. The species composition of the hydrolysate was analyzed using high efficiency anion exchange chromatography and UV spectroscopy. The result showed that the greatest balance between the residual solid and total dissolved solids was obtained at pH 4 and 170 °C. Maximum recovery rates of cellulose and lignin were as expected, whereas hemicellulose had the least recovery rate. The hemicellulose extraction rate was 42.19%, and the oligomer form accounted for 93.39% of the product. The physicochemical properties of bamboo with or without pretreatment was characterized. Compared with the traditional hydrothermal pretreatment, the new pretreatment bamboo has higher fiber crystallinity and thermal stability. In the pretreatment process, the fracture of β-aryl ether bond was inhibited and the structural dissociation of lignin was reduced. The physicochemical properties of bamboo was protected while the hemicellulose was extracted efficiently. It provides theoretical support for the efficient utilization of all components of woody biomass.

## 1. Introduction

Energy and environmental issues have become a global concern [1,2]. The Earth’s known reserves of coal, oil, and natural gas are being consumed. Therefore, an economic model based on fossil fuel-based energy is necessary to transform the economic model based on biomass (neutral carbon source) energy [3,4]. The idea of biomass refining resulted from these concerns. Profits from the paper industry are falling in the shrinking global economy along with reduced consumption of newsprint and increased costs of energy investments [5,6]. The pulp and paper industry is facing significant production pressure. Therefore, one of the most effective ways to diversify the product is to add increased value, eliminate the current predicaments, and increase the competitiveness of the pulp and paper industry. The pattern of the traditional pulping paper industry with pulp and paper as a single product has been altered in the biomass refining model based on biochemical and thermal chemical processes [7,8].

The general biorefinery concept focuses on the production of a variety of goods, including fuels, power, materials, and chemicals, from different biomass sources using integrated technologies [9,10]. Pretreatment has become the prerequisite and central unit of operation in the biorefinery process [11,12,13] to effectively utilize each component of lignocellulose biomass [14]. Hydrothermal pretreatment is an environmentally benign process that has been widely used for lignocellulose biomass pretreatment [15]. At present, the research on hydrothermal pretreatment focuses not only on wood materials, but also on bamboo and other non-wood materials [16,17]. The hemicellulose of bamboo was extracted in large quantities during the pretreatment, which effectively promoted the enzymatic hydrolysis efficiency [18]. However, during the hydrothermal pretreatment of bamboo, the degradation of cellulose increased with the dissolution of hemicellulose [17]. Many factors considerably influence the hydrothermal pretreatment of biomass, such as the physical properties of the biomass and the physical and chemical properties [19]. Other factors include the type of reactor and the catalyst. The hydroxyaldehyde condensation reaction can be easily achieved under alkaline conditions [20,21]. Acidic conditions were found to be suitable for dehydration [22]. Sulfuric acid and sodium hydroxide have been used to change the acid–base environment of the reaction. By adjusting the pH, the reaction direction can affect the direction to effective products [23]. High value utilization of woody biomass was promoted by hydrothermal pretreatment. For example, the separation and purification of cellulose were enhanced. The nanocellulose was prepared by oxidation of cellulose [24,25]. The application of nanocomposites was promoted by surface modification/surfactant polymerization of nanocellulose [26,27].

A novel, pre-corrected pH hydrothermal pretreatment was performed to integrate pH control without additional steps or equipment, resulting in the best balance between the residual solid and dissolved solid [3]. A buffer system was established by pre-correcting the pH value of the extraction solution. It neutralizes the hydronium ions in the extraction solution and stabilizes the pH of the extraction solution. Bamboo has the advantages of fast growth, a wide distribution area, and high yield; its fiber morphology and chemical composition have good pulping properties. Bamboo has the potential to be an important source of raw papermaking material by replacing broad-leaved wood. Indeed, bamboo is an important species for sustainable forestry development.

This paper primarily focuses on efficient pre-correcting of the pH for the hydrolysis efficiency of carbohydrates during hydrothermal pretreatment. The effects of pH and temperature on mass removal (MR) and total dissolved solids (TDS) were studied. In addition, the effect of pH on the hydrolysis mechanism of cellulose, hemicellulose, and lignin were studied. The species composition and purity of the hemicellulose and cellulose extracts were analyzed using high efficiency anion exchange chromatography (HPAEC). The content of phenolic compounds (PCs) formed from the degradation of lignin was analyzed via UV. The physicochemical structure of bamboo with or without pretreatment was characterized using XRD, 2D-NMR, TGA, and SEM.

## 2. Materials and Methods 

### 2.1. Materials

Bamboo was kindly supplied by a local bamboo forest (Guangxi, China): 2–3-year-old *bambusa chungii* McClure, cut sections of trunk. The chemical composition of the bagasse was analyzed by NREL method. First, 2 g of bamboo powder was added to 10 mL of 72% (*w*/*w*) H_2_SO_4_ for 7 min at 45 °C. The initial hydrolysis was interrupted by adding 50 mL of pure water and then diluted to 275 mL. The complete hydrolysis of oligomers was autoclaved for 30 min at 121 °C. The mixture was filtered. The hydrolysate was collected and diluted to 500 mL. The hydrolysate was analyzed via HPLC in a Waters e2695 (Waters, Milford, MA, USA) chromatograph equipped with a refractive index detector and symmetry shield TM RP18 column. The monosaccharide components and acetyl in the hydrolysate were analyzed. The solid obtained in the filtration after hydrolysis was oven-dried and weighed. The mass obtained corresponded to the residual lignin (Klason lignin), and the soluble lignin was determined by spectroscopy at 280 nm. The chemical composition of the bamboo was 22.68% lignin, 43.62% cellulose, 26.21% hemicellulose, and 2.97% acetyl. These analytical chemicals were purchased from Aladdin (Shanghai, China). All assay reagents were obtained from Sigma (St. Louis, MO, USA).

### 2.2. Hydrothermal Pretreatment

Hydrothermal pretreatment was carried out in a digester with six 1000-mL stainless steel cylinder reactors (Greenwood, Steuben, NY, USA). One hundred grams of bamboo chips (30 mm × 5 mm) were added to pure water, and the solid to liquid ratio was 1:5. The pH of the extraction solution was pre-corrected by adding 3.9 M sodium hydroxide and 4.5 M oxalic acid [3,4]. The mixture was then heated to a temperature of 145–170 °C for 90 min followed by thermal insulation for 60 min. The extraction liquid was quickly frozen and preserved after centrifugation. The residual solid (RS) and hydrolysate were collected.

### 2.3. Detection of MR and TDS Contents 

The difference between the raw material and RS is the mass removal (MR). TDS comprised the solid in the hydrolytic solution and was obtained by freeze-drying at −60 °C. It was primarily composed of nonvolatile soluble salts and organic compounds. The organic compounds were primarily hemicellulose, cellulose, and lignin.

### 2.4. RS Component Analysis

The composition of the RS after hydrolysis was detected using a standard method. The main components of the RS were found to be cellulose, hemicellulose, and lignin. The recovery rates of cellulose, hemicellulose, and lignin were calculated based on the RS component analysis. Some monosaccharide was hydrolyzed during hydrothermal pretreatment. The calculation of component recovery was affected by monosaccharide degradation. Correction of the monosaccharide degradation was modified by Li [28]. For example, the recovery ratio of cellulose was calculated according to Equation (1),
*Ri* = (*Ci* × *R_S_*)/*Mi*(1)
where *Ri* is the recovery of cellulose in %. *R_S_* is the recovery of RS in %. *Ci* is the content of cellulose in the *R_S_* (%). *Mi* is the content of cellulose in the raw material (%).

### 2.5. Detection of Sugars 

The hydrolytic solution contains different types of monosaccharides and polysaccharides mainly due to the different degrees of cellulose and hemicellulose degradation. However, the polysaccharide content could not be directly detected via ICS-5000+ HPAEC (Thermo Scientific Dionex, Sunnyvale, CA, USA). The polysaccharides were degraded into monosaccharides by further acid hydrolysis. The content of polysaccharide was the difference in the monosaccharide content with and without further acid hydrolysis. The polysaccharides in the hydrolytic solution were further degraded into monosaccharides using the NREL method [4,29]. The basic NREL method is as follows: the sample was mixed with 4.0% sulfuric acid solution, the mixture was heated to 121 °C for 70 min, and the sample was neutralized to pH 5–6 and diluted to the proper concentration. The diluted samples were filtered with a 0.22 μm filter.

The sugar component of the bamboo was detected using the NREL method. The basic method was as follows: 50 mg of bamboo powder was mixed with 1 mL of 72% sulfuric acid, the mixture was shaken at 25 °C for 1 h, and the sample was subsequently diluted with 17 mL of deionized water. The mixture was heated to 120 °C for 60 min.

The monosaccharide content in the hydrolytic solution was determined by HPAEC. The basic process is as follows. Deionized water and sodium acetate were used as the eluent, the flow rate of the eluent was 0.6 mL min^−1^. Then, 0.2 mol L^−1^ of sodium hydroxide was used as the regeneration agent at a flow rate of 0.6 mL min^−1^, and the supporting electrolyte was 0.5 mol L^−1^ sodium hydroxide at a flow rate of 1 mL min^−1^.

### 2.6. Acetic Acid and Furfural Detection

Acetyl is present in the branched chain of hemicellulose molecules. Acetic acid is produced by the hydrolysis of acetyl. In a hydrolysis solution, the acetyl exists in the form of acetic acid and acetyl oligomers. The content of acetic acid was detected by HPAEC in the hydrolytic solution. Acetyl oligomers were found to comprise the remaining part of the hydrolytic solution. Xylose was produced from the hydrolysis of hemicellulose. Furfural and 5-HMF were produced by the degradation of xylose and glucose. The content of furfural and 5-HMF in the hydrolysis solution was determined by HPAEC; the furfural and 5-HMF contents were then used to characterize the degradation of xylose and glucose.

### 2.7. Sugar Yield Calculation

Xylose, glucose, mannose, and other monosaccharides were produced by the hydrolysis of hemicellulose and cellulose in the hydrolysis process. The rate of dissolution cellulose and hemicellulose were investigated by determining the yield of monosaccharides. For example, the calculation for xylose is shown in Equation (2),
*Y* = [*C* × (*L*/*S*)] × 100%/*X*(2)
where *Y* is the yield of xylose in %. *C* is the concentration of xylose in the hydrolysate (g L^−1^). *L*/*S* is the liquid–solid ratio of the pre-extraction hot water at 5:1, and *X* is the content of xylose in the RS (g kg^−1^).

### 2.8. Semiquantitative Analysis of the PCs in the Hydrolysate

Phenolic compounds (PCs) were produced by the degradation of lignin during the hydrothermal pretreatment. Lignin has a special absorption peak at 280 nm in the UV spectrum. The degradation of lignin was detected by UV. The PCs were mainly derived from the hydrolysate and the cleaning solution of the RS. The total absorbance was calculated according to Equation (3),
*T* = *V*_1_ × *D*_1_ × *A*_1_ + *V*_2_ × *D*_2_ × *A*_2_(3)
where *T* is the total absorbance at 280 nm. *V*_1_ and *V*_2_ are the volumes of the hydrolysate and cleaning solution in mL, respectively. *D*_1_ and *D*_2_ are the dilution factors of the hydrolysate and cleaning solution in mL^−1^, respectively, and *A*_1_ and *A*_2_ are the absorbances of the PCs and washing water, respectively.

### 2.9. Physicochemical Structural Characterization

The bamboo powder was crushed and filtered using an 80-mesh sieve to obtain the powder. The crystallinity of the powder was determined using XRD (Mini Flex 600, Rigaku, Tokyo, Japan). A certain amount of wood powder was tiled into the substrate and flattened. The powder XRD spectrum was recorded using monochromatic Cu-Kα radiation with a voltage and current of 40 kV and 15 mA, respectively. The scan speed, scan range, and step size were 10 deg/min, 5–80°, and 0.01° [30].

The structural characteristics of the lignin were detected using 2D-heteronuclear single quantum coherence (HSQC) NMR spectroscopy [31]. The structural changes in the bleached and unbleached lignin were analyzed. The basic detection methods were as follows. First, 50 mg of EMAL was dissolved in 1 mL of DMSO-d6. The solvent peak of DMSO-d6 was used as a reference point for the internal chemical shift (δC/δH = 39.5/2.49). Additionally, ^1^H-^13^C correlation 2D-HSQC-NMR spectra were recorded at 25 °C on a Bruker DRX-500 spectrometer (Karlsruhe, Germany).

TGA (STA 449 F5 Jupiter, Netzsch, Selb, Germany) studies of the bamboo powder were carried out in an inert atmosphere of nitrogen with a purge rate of 40 mL min^−1^ using Al_2_O_3_ as the reference. The temperature was scanned from 25 °C to 800 °C with a heating rate of 10 °C min^−1^ [32].

The surface morphologies of the bamboo raw materials and samples that had been subjected to hydrothermal pretreatment were characterized using SEM (S-3400 N, Hitachi, Tokyo, Japan). The operating parameters of the SEM were set to extra high tension (EHT) (10 kV) with a working distance (WD) of 8.3 mm. The bamboo powder was sprayed with gold [3].

## 3. Results and Discussion

### 3.1. Effects of Operation Conditions

The effect of extraction can be quantified by the amount of MR in wood biomass materials [33]. The content of TDS in the hydrolytic solution is also an important index to measure the extraction effect [34]. TDS is also an important index to measure the degree of degradation of the extract in the hydrolytic solution. The effect of end pH and temperature on the MR and TDS during hydrothermal pretreatment was studied. The end pH values of the solutions were 3.0, 3.5, 4.0, 4.5, 5.0, and 5.5, respectively. The hydrolysis temperatures were 145 °C, 155 °C, and 170 °C, respectively. The end pH and temperature change during the experiment was within a reasonable range, and the industrial application of the hydrothermal pretreatment was not affected. The results are shown in Figure 1.

As shown in Figure 1a, the MS content increased with higher pH and temperature during hydrothermal pretreatment. At 145 °C, it showed a linear increase with an increase in pH when the pH was lower than 5.0. The MS content was 3.02% at pH 3.0 and increased to 13.23% at pH 5.0. However, the increasing trend of the MS content decreased when the pH was above 5.0. It was 14.81% at pH 5.5. At 155 °C, the MS content was 10.22% at pH 3.0, which increased to 17.61% at pH 5.0, and it was only 0.8% higher than that of pH 5 at pH 5.5. At an extraction temperature of 170 °C, the MS content was 18.81% at pH 3.0 and increased to 27.91% at pH 5.0. At pH 5.5, the MS content was only 0.3% higher than that at pH 5.0. The results showed that the extractable components of bamboo were nearly depleted with the pH of the hydrolysate increased at the same temperature. Thus, the MS content remained constant when the pH was further increased. And the higher temperatures led to the higher maximum solid removal at the same pH. This is mainly due to the increase of pH and temperature, which promote the dissociation of glycosidic bonds and β-aryl ether bonds [35]. The degradation and dissolution of hemicellulose and lignin were promoted.

The effect of pH and temperature on TDS during hydrothermal pretreatment of the bamboo chips is shown in Figure 1b. The TDS content increased with pH at 145 °C and was 1.91% at pH 3.0. It then increased to 9.31% at pH 5.5. This is due to the fact that most of the MS exist as TDS at low temperatures. At a lower pretreatment temperature, the TDS content increased linearly with an increase in pH. However, the pH has both positive and negative effects on the TDS content at high temperature. The TDS content first increased and then decreased with pH at 155 °C. It was 5.11% at pH 3.0 and increased to 14.13% at pH 5.5. At pH 5.5, the TDS content decreased by 4.2% compared to the decrease in TDS content at pH 5.0. In fact, MS increases with temperature, while further degradation of carbohydrate polymers and lignin macromolecules increases [36]. Water-soluble monosaccharides and acetic acid are produced from further degradation of TDS. At an extraction temperature of 170 °C, the TDS content was 13.01% at pH 3.0, and it increased to 18.54% at pH 4.0. It began to decrease as the pH continued to increase. At pH 4.5, it decreased by 1.7% compared to the decrease in total dissolved solids at pH 4.0. This indicated that the TDS underwent a high degree of degradation at high temperature and high pH [36]. A maximum content and minimum degradation of TDS was shown at pH 4.0 at 170 °C. The end pH of the extraction solution was 5.5 during the traditional hydrothermal pretreatment of the bamboo at 170 °C. At pH values of 4.0 and 5.5, the MS contents were 22.31% and 28.21%, and the TDS contents were 18.50% and 11.20%, respectively. The results showed that the MS content increased by 13.78% and the TDS content increased by 23.33% after the pre-corrected pH hydrothermal pretreatment (pH 4.0) of the bamboo compared to the case of traditional hydrothermal pretreatment (pH 3.5). These results show that the greatest balance between the RS and TDS (recycled carbohydrates) was achieved by pre-corrected pH hydrothermal pretreatment (pH 4.0).

### 3.2. Recovery of Carbohydrate Polymers and Lignin 

The ability to utilize RS after hydrothermal pretreatment mainly depends on its composition. The effect of pH on the recovery rate of carbohydrate polymers and lignin in RS was studied. The end pH of the hydrolysate was controlled between 3.0 and 5.5. The pretreatment temperature was 170 °C. Other pretreatment conditions remain unchanged. The results are shown in Figure 2.

The recovery ratios of cellulose, hemicellulose, and lignin in RS are shown in Figure 2. The recovery rates of cellulose and lignin increased first and then decreased with an increase in pH. The recovery ratio of cellulose was 84.41% at pH 3.0. It began to decline at a pH above 4.0. The maximum recovery rate of cellulose was 89.64% at pH 4.0. The changes of the lignin recovery coincided with the changes in cellulose recovery. The lignin recovery was 81.48% at pH 3.0 and began to decline when the pH was above 4. The maximum recovery ratio of lignin was 84.98% at pH 4.0. This indicated that the hydrolysis of cellulose and the degradation of lignin were most inhibited at pH 4.0 by 170 °C hydrothermal pretreatment. Contrary to the trends of cellulose and lignin recovery, the recovery rate of hemicellulose decreased with pH and then increased. It was 59.95% at pH 3.0 and decreases with increasing pH below pH 4.0. The minimum hemicellulose recovery ratio of 54% was obtained at pH 4.0, which indicated that the highest extraction rate of hemicellulose was 46%. The extraction rate of hemicellulose was 13% higher than that with traditional hydrothermal pretreatment. These results agree with Persson et al. and Song et al., who have been validated that the extraction of hemicellulose relies on a pH buffer about 4 [37,38]. A large amount of dissolved hemicellulose exists in the form of oligomers. And the local chemical environment of bamboo fiber was stable. Thus, pH is generally considered to be a key factor in the dissolution of hemicellulose. The dissolution of hemicellulose was promoted at pH 4.0. The yield of hemicellulose extract improved, and the hydrolysis of cellulose and lignin was inhibited. This was also the essential reason for the RS decrease and the TDS increase. Further processing of the biomass, such as that required in biorefineries or pulping and papermaking, was promoted by the pre-corrected pH hydrothermal pretreatment (pH 4.0).

### 3.3. Hydrolysis and Further Degradation of Hemicellulose

The main structure of hemicellulose consists of xylan, which is first hydrolyzed to oligosaccharide during hydrothermal pretreatment [39]. This oligosaccharide is hydrolyzed into xylose in the hydrolysate, and xylose is further degraded to furfural and other degradation products under acidic conditions. The hydrolysis mechanism of other sugars in hemicellulose (such as mannose, arabinose and galactose) have been studied [40]. The main degradation product of arabinose, galactose, and mannose is furfural 5-hydroxymethylfurfural (5-HMF). The effect of pH on the hemicellulose degradation products from the hydrolysate was studied at 170 °C. Other pretreatment conditions remained unchanged. The results are shown in Figure 3 and Table 1.

Figure 3 shows the contents of xylose, xylo-oligomers, and total xylose in the hydrolysates. The monomeric xylose content remained unchanged after it decreased with an increase in pH. It was 16.12 mmol L^−1^ at pH 3.0 and increased to 0.40% at pH 4.5. Then, it remained constant when the pH was above 4.5. The change in total xylose content went through three stages with the increase in pH. The total xylose content decreased slightly between pH 3.0 and 4.0. It was 38.30 mmol L^−1^ and 36.29 mmol L^−1^. The total xylose content decreased rapidly between pH 4.0 and 4.5 and was 14.99 mmol L^−1^ at pH 4.5. This was not conducive to the recycling of hemicellulose. The decreasing trend for the total xylose content was weakened when the pH was above 4.5, and it was 9.99 mmol L^−1^ at pH 5.5. The difference between the contents of total xylose and monomeric xylose was the amount of xylo-oligomers. The xylo-oligomers content increased rapidly between pH 3 and pH 4 from 22.18 mmol L^−1^ to 33.89 mmol L^−1^, respectively. However, it rapidly decreased between pH 4 and 4.5 and was 14.59 mmol L^−1^ at pH 4.5. The decreasing trend in the xylo-oligomers content slowed down as the pH continued to increase. This was mainly due to the hydrolysis of the xylo-oligomers into xylose, which was promoted by the decrease in the pre-hydrolysis pH. As the pH decreased, xylose was further degraded to furfural, increasing its concentration in the hydrolysates. However, the xylose in the oligomer form reached a maximum at pH 4.0. The total xylose content in the bamboo chip was 22.17% (expressed as xylose). The extraction rate of total xylose was 9.35% at pH 4.0. Thus, 42.19% of the total xylose was removed from the bamboo chips during hydrothermal pretreatment. The content of xylose was 6.61%, whereas the remainder was xylo-oligomers. The hemicellulose extraction rates were 42.19% and 93.39% in the oligomer form at pH 4.0 under a 170 °C pre-corrected pH hydrothermal pretreatment. The results confirmed the promoting effect of pH 4 buffer system on hemicellulosic extraction [37,38].

The results of pH for the xylose saccharides and further degradation products are shown in Table 1. The pre-corrected pH had a significant effect on the hydrothermal pretreatment. The changes in arabinose and galactose were similar to those for xylose. At pH 4.0, the total arabinose and galactose saccharides produced were 0.19% and 0.08%, respectively, with the majority (0.11% and 0.08%) in the oligomeric form. With a decrease in pH, the content of furfural increased, owing to further degradation of xylose. The furfural production was not only caused by the degradation of five-carbon sugars but indicated that other chemicals, including furfural-based polymers, were involved in the hydrolysis reaction [41]. The content of acetyl decreased with an increase in pH and was 2.11% at pH 3.0. However, the acetyl group bound to oligomers was not detected. This result was mainly due to the presence of lipids, such as free fatty acids, glycerin, wax, and sterol [42]. In a strongly acidic environment, acetic acid reacts with these lipids. Therefore, the content of total acetic acid detected after secondary hydrolysis remained unchanged.

### 3.4. Hydrolysis and Further Degradation of Cellulose

Cellulose is the largest proportion of biomass [43]. Cellulose is used as a model compound to study the hydrolysis mechanism of complex biomass during hydrothermal pretreatment. One study found that cellulose undergoes a similar reaction mechanism via acid hydrolysis during hydrothermal pretreatment [44]. The effect of pH on the cellulose degradation products in the hydrolysate was studied at 170 °C. Other pretreatment conditions remained unchanged. The results are shown in Figure 4.

In Figure 4, the content of glucose, gluco-oligomers, and total glucose in the hydrolysate are shown. The total glucose content increased gradually with pH between 3.0 and 5.0, and was 3.84% at pH 3.0. It then increased to 15.21% at pH 5.0. The total glucose content in the hydrolysate remained unchanged after the pH was above 5. The glucose oligomer content increased from 3.33% to 5.09% at pH 3.0 to 3.5, respectively; however, there was a smaller increase in the glucose oligomer content between pH 3.5 and 4.0, which was 5.49%. It increased rapidly after the pH was above 4.0. The total glucose content and glucose oligomers increased with increasing pH. However, the content of glucose monomers increased first and then decreased with an increase in pH. The maximum glucose monomer content was 3.17 mmol L^−1^ at pH 4.0. The hydrolysis of gluco-oligomer into glucose was facilitated by the decrease in pH of the hydrolytic solution. The monomeric glucose content reached a maximum at pH 4.0. Compared to traditional hydrothermal pretreatment (pH 3.5), the contents of total glucose and gluco-oligomer remained unchanged during pre-corrected pH hydrothermal pretreatment (pH 4.0). Thus, the dissolution and degradation of cellulose was inhibited. As the pH continually decreased, 5-HMF was produced by degradation of glucose. The 5-HMF content decreased with increasing pH (Table 1) and was 5.02% at pH 3.0. It then decreased to 0.13% at pH 5.5. Cellulose hydrolysis was promoted when the pH was below 4.0 (Figure 2). The degradation of the glucose oligomers and monomeric glucose into 5-HMF was promoted by the low pH. At pH 4.0, the total glucose content was low, and most of the glucose was in the form of oligomers. This means that cellulose dissolution was most inhibited, and its degradation was minimal when hemicellulose was significantly extracted at pH 4.0 and 170 °C.

The crystallinity of bamboo fibers with or without hydrothermal pretreatment was studied by XRD. The results are shown in Figure 5. The crystallinity of the bamboo raw materials was 45.71%, which increased to 48.13% by hydrothermal treatment. The main reason was that hemicellulose, lignin, and other non-crystalline components were removed during the hydrothermal pretreatment. The proportion of cellulose increased [45]. The crystallinity of bamboo was 51.41% by pre-corrected pH hydrothermal pretreatment. The results indicate that the amorphous fraction in bamboo was significantly damaged by pre-corrected pH hydrothermal pretreatment. This was attributed to a high dissolution rate of hemicellulose and the slight degradation of cellulose, both of which represented amorphous components. However, traditional hydrothermal pretreatment was less selective regarding the degradation and dissolution of the main components of bamboo fiber. When hemicellulose was extracted, cellulose and lignin were degraded and dissolved, increasing the relative content of crystalline cellulose in bamboo. Thus, the crystallinity increased slightly by hydrothermal pretreatment.

### 3.5. Changes in the Lignin Content and Structure

The content of PCs in the hydrolytic solution is an important index used to measure the dissolution and degradation of lignin [46]. For hydrothermal pretreatment, the dissolved lignin has less research significance, but the content of aromatic compounds is an important parameter because the recovery of products and the purification of sugar are affected by the aromatic compounds. Thus, the dissolution of PCs was examined. The effect of pH on the dissolution and deposition of lignin was studied at 170 °C. Other hydrothermal pretreatment conditions remained unchanged. The results are shown in Figure 6.

Figure 6 shows the change in PC content in the hydrolysates as a function of pH. The content of phenolic compounds first increased and then decreased; it was 5.70% at pH 3.0 but rapidly decreased to 2.45% at pH 4.0. The content of PCs increased slowly when the pH was above 4.0, and was 3.03% at pH 5.5. At pH 4.0, the content of PCs in the extraction liquor was the lowest. This means that the smallest amount of lignin was dissolved and degraded. This is due to the suppression of the fracture of the β-aryl ether bond. The structural dissociation of lignin was inhibited (see below). The content of PCs decreased by 33.78% during pre-corrected pH hydrothermal pretreatment (pH 4.0) compared to that during traditional hydrothermal pretreatment (pH 3.5). Thus, the dissolution and degradation of lignin was maximally inhibited, while hemicellulose was heavily extracted at pH 4.0 by 170 °C pre-corrected pH hydrothermal pretreatment.

The structural changes in the lignin in RS with and without hydrothermal pretreatment were studied, and the results are shown in Figure 7. Figure 7a shows the structural characteristics of lignin from raw materials. Strong signals representing methoxyl group (OMe), C_α_-H_α_, C_β_-H_β_, and C_γ_-H_γ_ correlations in the β-O-4 substructures were observed. This indicated that β-O-4 was the major inter-unit of the lignin structure. The C_β_-H_β_ and C_γ_-H_γ_ correlations in the resinol substructures (B) were observed at δ_C_/δ_H_ 53.63/3.06, 70.05/3.82, and 4.18 ppm. The C_α_-H_α_, C_β_-H_β_, and C_γ_-H_γ_ correlations in the phenylcoumaran substructures (C) were observed at δC/δH 86.83/5.45, 53.03/3.46, and 62.5/3.72 ppm. In addition to the phenyl coumarone (C) condensation structure of lignin, spiraldienone (D) was also found in the side chain. The aromatic regions of bamboo lignin were mainly composed of three basic units: guaiacyl (G), syringyl (S), and p-hydroxyphenyl (H) [47]. C_2,6_-H_2,6_ correlations in the S_0_ units were observed at δ_C_/δ_H_ 103.8/6.71 ppm. The chemical shift of S′_2,6_ of the syringyl group with carbonyl in C_α_ was at δ_C_/δ_H_ 106.33/7.32 ppm. The C_2_-H_2_, C_5_-H_5_, and C_6_-H_6_ correlations in the G-lignin were observed at δ_C_/δ_H_ 110.9/6.99, 115.63/6.78, and 118.86/6.83 ppm, respectively. There were also p-coumarinate residues (PCA) in the raw lignin. C_α_−H_α_, C_β_−H_β_, C_2_-H_2_, and C_6_-H_6_ were observed at δ_C_/δ_H_ 144.7/7.42, 113.7/6.27, and 130.1/7.46 ppm, respectively. H-lignin mainly form ester bonds with the side chain γ-hydroxyl groups of β-O-4 in the form of the p-coumarin acetyl group [48,49,50].

Figure 7b shows the 2D-HSQC spectra of bamboo lignin by traditional hydrothermal pretreatment. The signals for A_β_(S) and A′_β_ were weakened, indicating that the β-O-4 bond of lignin was broken during hydrothermal pretreatment. The signal for S′_2,6_ with the carbonyl group at the C_α_ position was weakened. Thus, the α-carbonyl group of lignin was easily oxidized and degraded. The signal representing the G-lignin was weakened, indicating that the benzene ring structure of lignin was broken. In addition, the signal representing p-coumarin acetyl was weakened, indicating that p-coumarin acetyl degraded during hydrothermal pretreatment. The signal strength of the lignin changed after pre-corrected pH hydrothermal pretreatment (Figure 7c). The signals for A_β_(S) and A’_β_ were enhanced compared to those by traditional hydrothermal pretreatment. This is due to the degradation of S-lignin and the breaking of the β-O-4 bond were inhibited. The signal for C_5_-H_5_ in the G-lignin was enhanced. This also indicated that the dissolution and degradation of G-lignin were inhibited. The enhancement of the C_β_ signal indicated that the β-5 bond was stable and not easily destroyed. In fact, G-lignin, S-lignin, and associated bonds were protected during the pre-corrected pH hydrothermal pretreatment. The changes in lignin content above were verified by the transformation of lignin structure. The results showed that the degradation of lignin decreased, whereas more hemicellulose extracts were obtained by pre-corrected pH hydrothermal pretreatment.

### 3.6. Thermal Stability Analysis

Hemicellulose is typically the easiest to decompose. Cellulose pyrolysis generally occurs at higher temperatures. When the temperature was above 400 °C, almost all cellulose was decomposed completely, while lignin is typically the most difficult to decompose [48]. The effect of hydrothermal pretreatment on the thermal stability of bamboo was studied, and the results are shown in Figure 8.

Figure 8a shows the TG curves for the bamboo raw materials for hydrothermal pretreatment and hydrothermal pretreatment with the pre-corrected pH hydrothermal pretreatment. The initial degradation temperature of the raw materials was 282.64 °C, which increased to 298.91 °C and 313.44 °C with hydrothermal pretreatment and pre-corrected pH hydrothermal pretreatment, respectively. This was mainly due to the large amount of hemicellulose that was extracted. The rapid mass losses were 52.00%, 51.39%, and 46.51%, respectively. The total mass losses were 70.22%, 76.36%, and 74.66%, respectively. This was due to the presence of hemicellulose in bamboo, which can significantly reduce the initial temperature of rapid mass losses of fiber. With the removal of hemicellulose by pretreatment, the initial temperature of rapid mass losses of lignocellulosic fibers increased. Based on the extraction efficiency of hemicellulose, bamboo has a higher initial degradation temperature after hydrothermal pretreatment. It is well known that, in addition to hemicellulose being extracted in large quantities, a small amount of cellulose was dissolved and degraded during hydrothermal pretreatment. This leads to a decrease in rapid mass losses after hydrothermal pretreatment. In particular, the dissolution and degradation of cellulose were inhibited during pre-corrected pH hydrothermal pretreatment, resulting in a significant decrease in the rapid mass losses. The total mass losses of bamboo depends on the residual lignin content. A small amount of lignin was dissolved and degraded with the fracture of the β-aryl ether bond during hydrothermal pretreatment. This results in the increase of the total mass losses after hydrothermal pretreatment. However, it was slightly reduced for the inhibition of lignin dissolution and degradation in pre-corrected pH hydrothermal pretreatment.

The extraction of hemicellulose also resulted in the disappearance of the “acromion” peak after hydrothermal pretreatment [51] (Figure 8b). The maximum degradation temperatures of the materials were 334.96 °C, 347.39 °C, and 356.57 °C. The maximum mass loss rates were 8.19 min^−1^, 9.88 min^−1^, and 10.23 min^−1^. Compared to the traditional hydrothermal pretreatment, the maximum degradation temperature and the maximum weight loss rate of bamboo upon the pre-corrected pH hydrothermal pretreatment increased. This was ascribed to the removal of hemicellulose, whereas most of the cellulose and lignin were not affected, increasing the pyrolysis rate. In fact, the process of hydrothermal pretreatment provides a clean and efficient separation of the main components of bamboo, thereby promoting the pyrolysis reaction of bamboo.

### 3.7. Morphology Analysis

The effects on the surface morphology of bamboo with or without hydrothermal pretreatment were analyzed using SEM. The results are shown in Figure 9.

Figure 9a shows a large amount of impurity fragments adhering to the surface of the bamboo raw materials. These fragments are mainly composed of waxy [52], pectin [53] and other non-wood fiber main components. The surface of a single fiber bundle was covered with a large amount of impurities, as shown in Figure 9j, and the cross-linking between the fiber bundles was serious and disordered (Figure 9d). The fragments were stripped during hydrothermal pretreatment. However, the surface of the pretreatment bamboo was not smooth but cracked (Figure 9b). The fiber bundles remained relatively intact (Figure 9e), but the surface of a single fiber bundle was damaged (Figure 9h), which was mainly due to the weak acid erosion during hydrolysis. Figure 9c shows that the fragments were stripped thoroughly, and the bamboo surface was smooth after pre-corrected pH hydrothermal pretreatment. The fiber bundles were clearly distributed and independent (Figure 9f). The surface of the single fiber bundle was smooth (Figure 9i). A large amount of hemicellulose and a small amount of lignin were extracted during hydrothermal pretreatment. The intact bamboo fiber network was destroyed [54]. The surface of the fiber bundle became rough [55]. The dissolution and degradation of cellulose and lignin were inhibited, while hemicellulose was extracted in large quantities during pre-corrected pH hydrothermal pretreatment. The damage to the fiber bundles was reduced, and the surface of each single fiber bundle was complete and smooth. The results showed that the hydrothermal pretreatment had less effect on the physicochemical properties of bamboo, but had higher extraction efficiency of hemicellulose.

## 4. Conclusions

Considerable application prospects for pre-corrected pH hydrothermal pretreatment in biorefineries have been demonstrated. Reasonable residual solids and maximal total dissolved solids can be expected. A high yield and a high molecular weight for hemicellulose were obtained by pre-corrected pH hydrothermal pretreatment. The dissolution and degradation of cellulose and lignin were inhibited. The structural dissociation of lignin was controlled. Higher fiber crystallinity, better thermal stability, and less surface erosion were obtained. The new method is beneficial to the efficient cleaning and separation of main components of woody biomass and biorefineries.

## Figures and Tables

**Figure 1 polymers-12-00612-f001:**
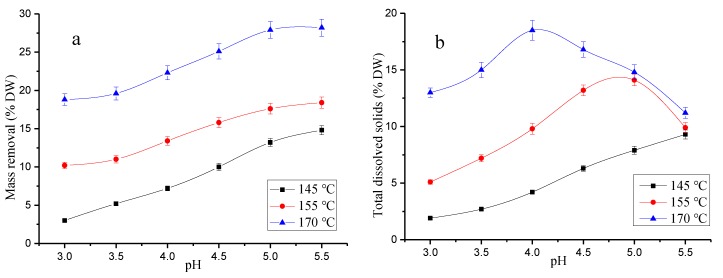
Effect of pH and temperature on MS and TDS in hydrothermal pretreatment (**a**) MS removal, (**b**) TDS content.

**Figure 2 polymers-12-00612-f002:**
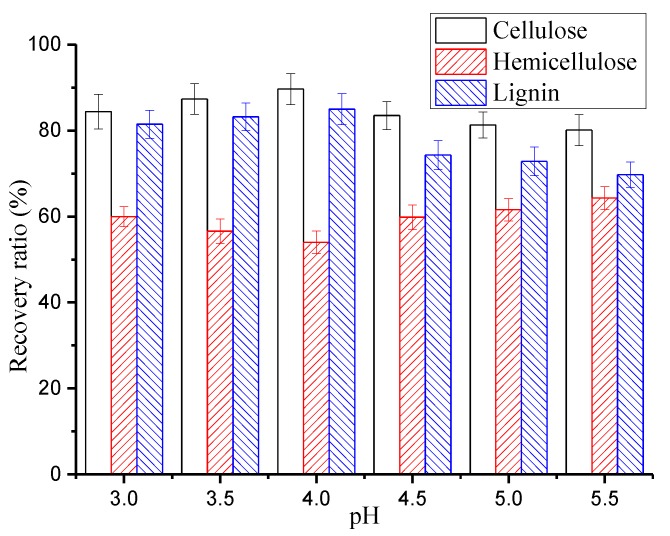
Effect of hydrothermal pretreatment on recovery ratio.

**Figure 3 polymers-12-00612-f003:**
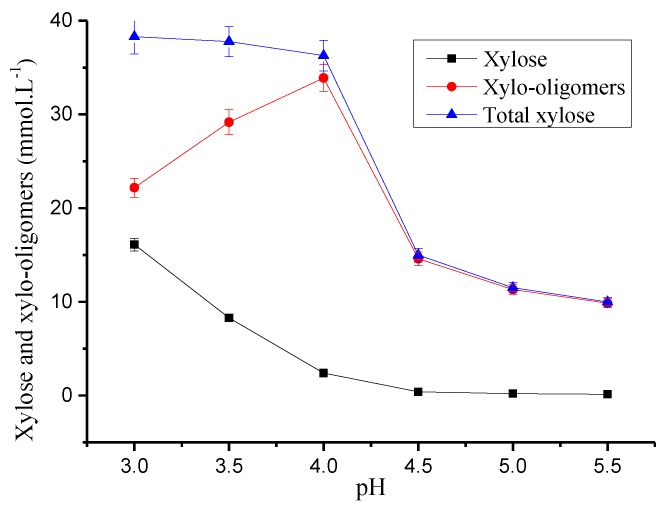
Effect of pH on xylo units in pre-corrected pH hydrothermal pretreatment.

**Figure 4 polymers-12-00612-f004:**
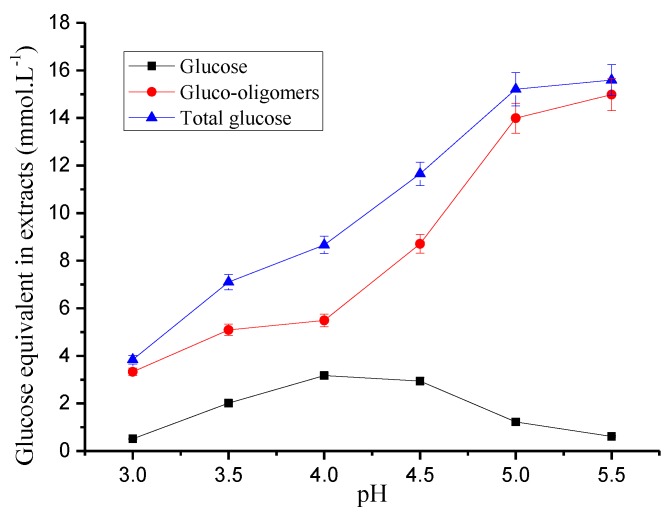
Effect of pH on glucose-units in pre-corrected pH hydrothermal pretreatment.

**Figure 5 polymers-12-00612-f005:**
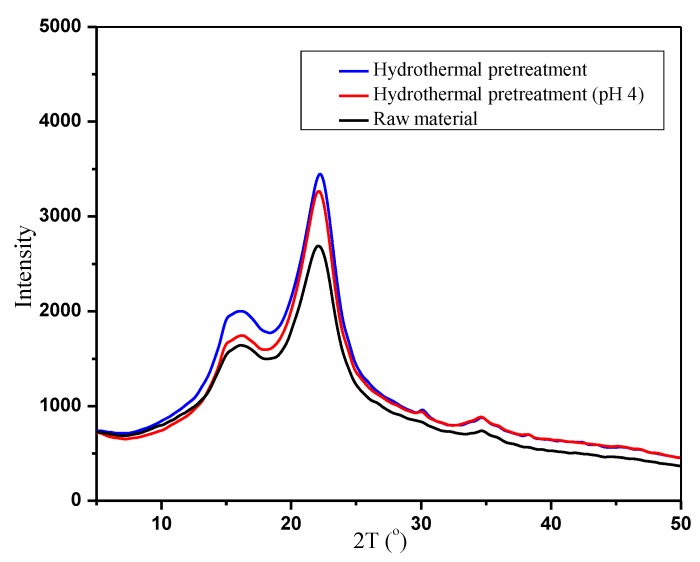
XRD spectra of bamboo with or without hydrothermal pretreatment.

**Figure 6 polymers-12-00612-f006:**
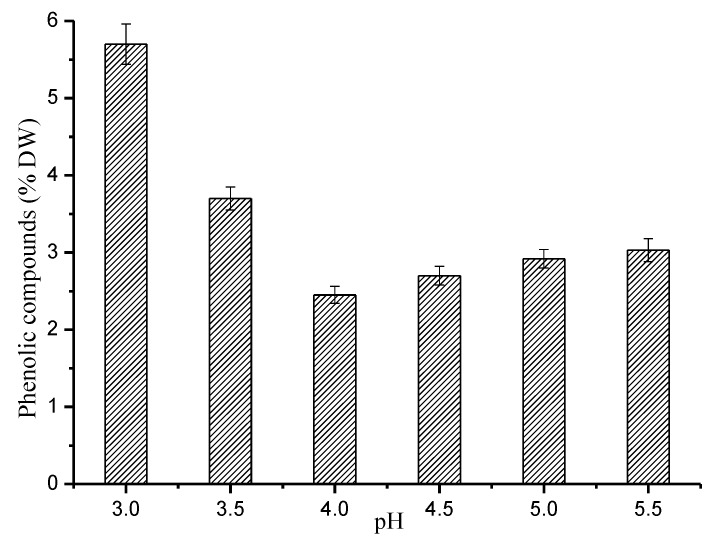
Effect of pH on PCs in pre-corrected pH hydrothermal pretreatment.

**Figure 7 polymers-12-00612-f007:**
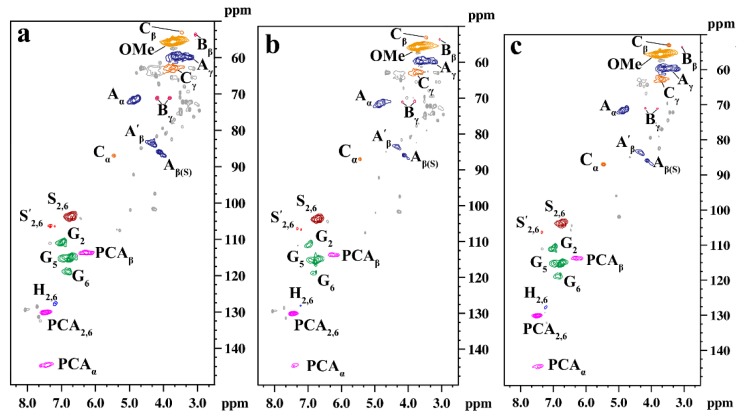
2D-NMR of lignin with and without pretreatment (**a**. raw material, **b**. traditional hydrothermal pretreatment, **c**. pre-corrected pH hydrothermal pretreatment).

**Figure 8 polymers-12-00612-f008:**
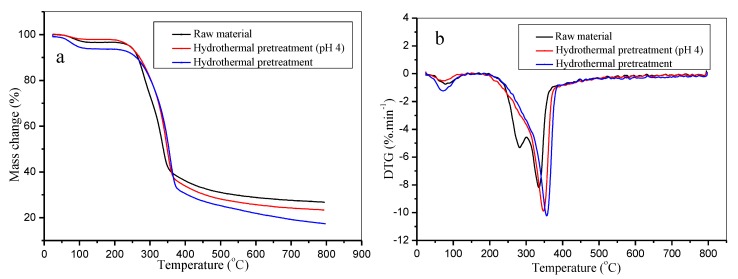
Thermal stability of bamboo with or without pretreatment (**a**. TGA curves, **b**. DTG curves).

**Figure 9 polymers-12-00612-f009:**
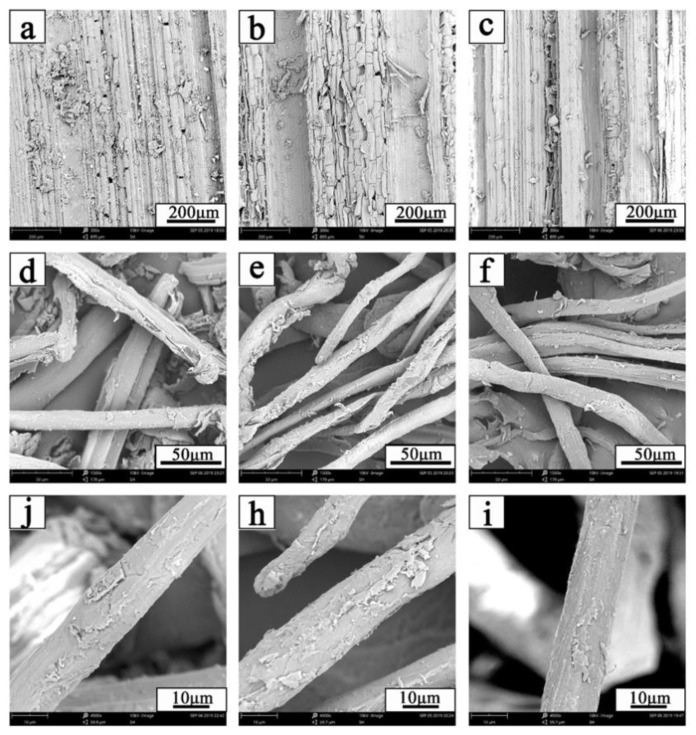
SEM of bamboo with or without hydrothermal pretreatment (**a**,**d**,**j**): raw material. (**b**,**e**,**h**): traditional hydrothermal pretreatment. (**c**,**f**,**i**): pre-corrected pH hydrothermal pretreatment).

**Table 1 polymers-12-00612-t001:** Chemical composition of hydrolysates at different pH value (based on DW, %).

**pH**	3.0	3.5	4.0	4.5	5.0	5.5
**Arabinose**						
**Monomeric**	0.18 ± 0.009	0.16 ± 0.008	0.08 ± 0.004	0.02 ± 0.001	0.02 ± 0.001	0.02 ± 0.001
**Oligomeric**	0.03 ± 0.001	0.04 ± 0.002	0.11 ± 0.005	0.08 ± 0.003	0.06 ± 0.003	0.06 ± 0.003
**Galactose**						
**Monomeric**	0.07 ± 0.003	0.05 ± 0.002	0.00	0.00	0.00	0.00
**Oligomeric**	0.02 ± 0.001	0.03 ± 0.001	0.08 ± 0.004	0.05 ± 0.002	0.05 ± 0.002	0.05 ± 0.002
**Acetyl**	2.11 ± 0.095	0.91 ± 0.041	0.35 ± 0.017	0.12 ± 0.006	0.08 ± 0.004	0.04 ± 0.002
**Furfural**	3.40 ± 0.153	1.83 ± 0.082	0.55 ± 0.026	0.32 ± 0.014	0.12 ± 0.006	0.07 ± 0.003
**5-HMF**	5.02 ± 0.226	2.84 ± 0.119	1.07 ± 0.051	0.43 ± 0.018	0.19 ± 0.008	0.13 ± 0.006

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
