# Peer review of "Effect of Pre-Corrected pH on the Carbohydrate Hydrolysis of Bamboo during Hydrothermal Pretreatment"

_polymers, 2020, doi:10.3390/polym12030612_

Round 1

Reviewer 1 Report

The manuscript presents a study on the use of hydrothermal treatment for bamboo biomass. The study contributes to a growing area of research on bamboo biomass and methods of pre-treatment to increase yields. The following comments are provided to the authors for consideration:

L70 – The authors use the term “bagasse” in reference to the bamboo material tested which is incorrect as it refers the residual pulp from sugarcane production. Please revise.

L70 – Please clarify the form of the bamboo material obtained from the local forest, was the raw material the full bamboo culm, cut sections (e.g. top, middle, bottom), residual products from processing, etc. These details along with age and species are important to understanding the results of the study.

L70 – Please state the standards used for the methods described.

L90 – Please define MR.

L112 – Please provide and cite the NREL standard and reference.

L144 – Please define PC.

L165 – Please clarify why wood powder was used.

L347 – The resolution and visibility of the labels Figure 6 should be improved to allow for clear view of the comparison presented in the discussion.

Reviewer 2 Report

This manuscript describes the effect of “pre-corrected” pH on the carbohydrate hydrolysis of bamboo during hydrothermal pretreatment. It is logically constructed but some revisions are required.

Introduction:

There is a need to discuss properly previous studies on hydrothermal pretreatment of bamboo

Materials and Methods:

 The bamboo specie used should be specified.  Bamboo chip size and size distributions should also been specified as they have a major effect on the hydrolysis.

Reference #4 was cited for the method of “pre-correcting” pH with sodium hydroxide and oxalic acid. However, this reviewer cannot find any information regarding to oxalic acid addition in the reference #4. A detail description of the experimental protocols is needed.

Lines 180-182

The final pH should be specified for each of experiment to evaluate the effectiveness of the buffer system applied. For industry application, hydrolysis end pH can be effectively controlled using a feedback control approach anyway. This reviewer is not convinced that there is enough date to support the merits of a buffer system with sodium hydroxide and oxalic acid for industrial application.

Reviewer 3 Report

The authors submitted the article entitled “Effect of pre-corrected pH on the carbohydrate hydrolysis of bamboo during hydrothermal pretreatment”. I would recommend that the paper could be published elsewhere. My main comments and questions are as follows:
1. The article is like a technical report. It does not match the scope of this journal.
2. Overall, the motivation is trivial. The only discussion of hydrothermal and pH treatment toward cellulose cannot match the scope of the journal.
3. Although the authors provide much data, I can’t see any fundamental or basic discussion.
4. The authors should provide some microscope results to detect the morphology of the resultants.
5. The authors should also provide the XRD data of the resultants
6. The authors should check the format of the references.
7. English correction is recommended since there are many unclear descriptions.

Round 2

Reviewer 3 Report

The authors have revised the manuscript extensively.

One point  is that the authors should cite/review some recent cellulose-related papers in POLYMERS or else, such as from the oxidation of cellulose to nanocellulose and its surface modifications/surface living polymerizations towards the applications of nanocomposites to enrich the manuscript value.

Author Response

We sincerely thank you for spending considerable time and efforts in reviewing this manuscript. We have made the effort to respond to all the comments and have revised the manuscript according to the reviewers' suggestions. All altered contents have been marked in the revised manuscript. A point-by-point response to the reviewer’ comments is as follows.

Point 1: One point  is that the authors should cite/review some recent cellulose-related papers in POLYMERS or else, such as from the oxidation of cellulose to nanocellulose and its surface modifications/surface living polymerizations towards the applications of nanocomposites to enrich the manuscript value.

Response 1: Some recent cellulose-related papers in POLYMERS or CARBOHYDRATE POLYMERS had been cited. It mainly involves the preparation of nanocellulose and the modification of nanocomposites. As a whole, the comment is very pertinent, which are very helpful to modify my entire paper and thank you very much again.